# Compressing LLMs: The Truth is Rarely Pure and Never Simple

 Ajay Jaiswal[1], Zhe Gan[2], Xianzhi Du[2], Bowen Zhang[2], Zhangyang Wang[1], Yinfei Yang[2]
[1]University of Texas at Austin, [2]Apple

## Abstract

Despite their remarkable achievements, modern Large Language Models (LLMs) face exorbitant computational and memory footprints. Recently, several works have shown significant success in *training-free* and *data-free* compression (pruning and quantization) of LLMs that achieve 50 - 60% sparsity and reduce the bit width to 3 or 4 bits per weight, with negligible degradation of perplexity over the uncompressed baseline. As recent research efforts are focused on developing increasingly sophisticated compression methods, our work takes a step back and re-evaluates the effectiveness of existing SoTA compression methods, which rely on a fairly simple and widely questioned metric, perplexity (even for dense LLMs). We introduce **K**nowledge-**I**ntensive **C**ompressed LLM Benchmar**K** (**LLM-KICK**), a collection of carefully curated tasks to redefine the evaluation protocol for compressed LLMs, which have significant alignment with their dense counterparts and perplexity fail to capture subtle change in their true capabilities. LLM-KICK unveils many favorable merits and unfortunate plights of current SoTA compression methods: all pruning methods suffer significant performance degradation, sometimes at trivial sparsity ratios (*e.g.*, 25-30%), and fail for N:M sparsity in knowledge-intensive tasks; current quantization methods are more successful than pruning; yet, pruned LLMs even at $\geq$ 50% sparsity are robust in-context retrieval and summarization systems; among others. LLM-KICK is designed to holistically access compressed LLMs' ability for language understanding, reasoning, generation, in-context retrieval, in-context summarization, *etc.* We hope our study can foster the development of better LLM compression methods. The reproduced codes are available at https://github.com/VITA-Group/llm-kick.

## 1 Introduction

Large Language Models (LLMs) are *omnipresent*, profoundly influencing not only the landscape of NLP (Ram et al., 2023; Liu et al., 2023a; Sawada et al., 2023; Qin et al., 2023; Zhuo, 2023; Lee et al., 2023), but also recently buttressing numerous computer vision (Lian et al., 2023; Wang et al., 2023; Lai et al., 2023; Lu et al., 2023) and graph neural networks (Ye et al., 2023; Chen et al., 2023; Qian et al., 2023; Duan et al., 2023) algorithms; achieving steller performance across various task leaderboards. Despite their numerous unprecedented capabilities, their democratization is primarily restricted by the presence of billions of parameters, which depends on astonishingly high computational and memory requirements. For example, GPT-175B requires 325 GB of GPU memory simply to load its model weights, and at least five A100 (80GB) GPUs with sophisticated parallelism techniques (Sheng et al., 2023).

To democratize LLMs, considerable efforts have been taking to mitigate their high computational cost, mainly divided into two research directions: *network pruning*, and *weight quantization*. The former shrinks network sizes by removing specific weights from the model – essentially setting them to zero, while the latter aims to quantize parameters into lower bit-level representations. Several recent success in network pruning (Sun et al., 2023; Frantar & Alistarh, 2023; Jaiswal et al., 2023a; Ma et al., 2023; Ji et al., 2023) and quantization (Liu et al., 2023c; Kim et al., 2023; Dettmers et al., 2023a; Frantar et al., 2022; Lin et al., 2023a; Dettmers et al., 2023c) (detailed related work

---

 Work done during an internship at Apple.

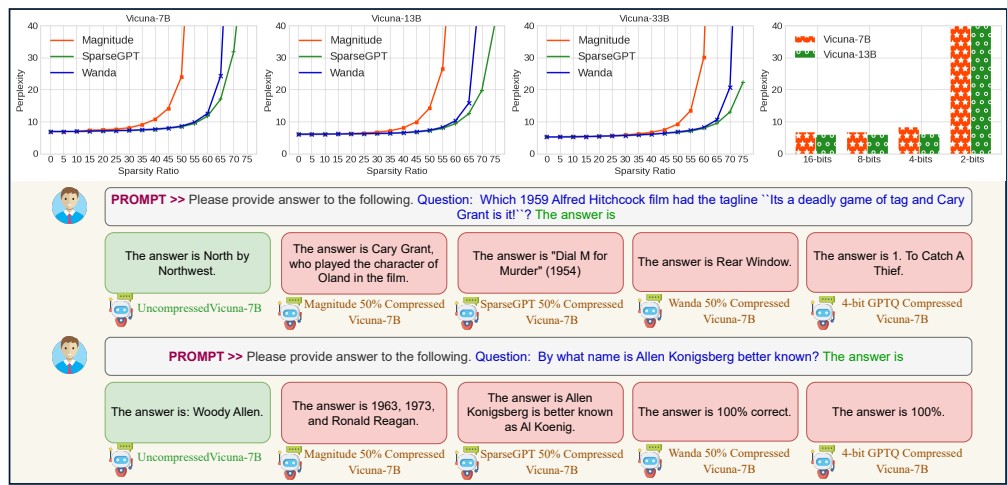

Figure 1: **True Merits of SoTA Compression.** Top row indicates marginal increase in perplexity via using SoTA compression methods, when compared with simple magnitude-based pruning. Bottom row indicates the failure of compressed Vicuna-7B (Chiang et al., 2023) (via Magnitude, Wanda, SparseGPT, GPTQ) to respond correctly to knowledge-intensive factoid-based questions.

discussion in Appendix A.1) claim to retain the uncompressed LLM's performance while achieving 50-60% sparsity or up to extreme 2-3 bit quantization. Although these advancements look fascinating, in most (if not all) cases, they heavily rely on **perplexity** as their primary metric to evaluate the performance claims. Such relatively restricted evaluations limit the scope for developing new compression methods, and are potentially ill-suited to identifying new and unexpected capabilities/limitations of compressed LLMs.

Perplexity, even in the case of dense LLMs, has been questioned as an unsatisfactory measure for comparing the true potential of LLMs, despite significant variations in model scales, training strategies, and architecture choices (Muhlgay et al., 2023). It is important to note that *all compressed models are derived from the same dense counterpart with high similarity*, and aforementioned differences don't exist, making their evaluation more challenging. In this work, we revisit a widely known yet under-explored question: *How well does perplexity capture the change in capabilities of compressed LLMs that have significant alignment with their dense counterpart?* We focus on the case of compressed LLMs, because we observe comparatively more serious failure of perplexity to capture the delicate performance variations incurred across varying compression stages of LLMs, demanding a more fine-grained investigation.

In this work, we attempt to investigate the true promises and limitations of state-of-the-art compression algorithms for LLMs. We assemble the **first** comprehensive and diverse collection of tasks with varying difficulty levels to thoroughly study compressed LLMs under quantization and network pruning (structured and unstructured sparsity patterns). More specifically, we consider a broad range of tasks to evaluate subtle changes in pruned and quantized LLMs' ability for *language understanding, reasoning, generation, in-context retrieval, long-context summarization*, *etc*. Note that none of the datasets in our multi-dimensional study of compressed LLMs was created from scratch, but we rely on existing datasets as they have been widely accepted by researchers, but unfortunately yet not been adopted to study the effect of compression. We rigorously measure the performance of SoTA quantization and pruning approaches (in their most common, default settings), to understand their potential for our challenging and interesting tasks with high practical value.

Our key observations and contributions can be unfolded as:

- We present *Knowledge-Intensive Compressed LLM BenchmarK* (**LLM-KICK**), to **re-define** the evaluation protocols for compressed LLMs and facilitate a comprehensive assessment of SoTA compression algorithms. The premise of our work is to develop a suite of challenging, realistic, and diverse tasks of high practical importance and datasets that can empower a systematic understanding of how *existing LLM compression strategies* truly perform in preserving performance

despite their similar perplexities, how they differ from each other, and how they compare against smaller LLMs of comparable parameter counts.

- LLM-KICK unveils many interesting and critical observations, that perplexity-based evaluations overlook. ① Most SoTA pruning methods suffer significant performance degradation, sometimes *at trivial sparsity ratios (e.g., 25-30%)*, despite negligible changes in perplexity. ② All SoTA pruning methods do not work satisfactorily for structured N:M sparsity patterns on LLM-KICK. ③ Current SoTA LLM quantization methods are *more successful* in perpetuating performance in comparison to SoTA LLM pruning methods. ④ Compressed LLMs fail to generate knowledge-enriched and factually correct answers, despite the generated text is fluent, consistent, and coherent. ⑤ Compressed LLMs with larger architectures but same parameter counts perform poorer, which favors smaller dense models.

- We further investigate compressed LLMs' ability for *in-context settings*, via adopting *in-context retrieval augmented question answering* (ICRA-QA) (Ram et al., 2023), and *text summarization with in-context learning* (IC-Sum) (Jain et al., 2023). To our surprise, pruned LLMs, even at non-trivial sparsity ratios (*e.g.*, ≥50%), are **robust** retrieval systems, and can perform text summarization while maintaining similar performance as their dense counterpart. However, with increasing compression degrees, their ability to digest longer context is **affected more** than smaller context.

## 2 SoTA LLM Compression: Perplexity, or What's More?

Scaling neural networks, now LLMs, have achieved astonishing performance benefits on a wide array of tasks, but at the cost of gigantic computational and memory footprints. Network pruning and weight quantization are two popular remedies to mitigate these overheads due to billions of parameter counts in current LLMs. Despite numerous existing algorithms for pruning (Singh & Alistarh, 2020; Zhu & Gupta, 2017; Gale et al., 2019; Jaiswal et al., 2022; Lin et al., 2020; Liu et al., 2021a; Mostafa & Wang, 2019; Dettmers & Zettlemoyer, 2019; Evci et al., 2020) and quantization (Dong et al., 2022; Cardinaux et al., 2020; Kim et al., 2021; Liu et al., 2021b; Martinez et al., 2020), their ad-hoc adaptation for LLMs is restricted, due to the lack of luxury to perform iterative re-training to regain any performance drop during compression. Recently, several works have shown significant success in training-free and data-free compression of LLMs achieving 50-60% sparsity and reducing the bit-width down to 3 or 4 bits per weight, with negligible perplexity degradation relative to the uncompressed baseline.

Perplexity is a statistical measure of how confident a language model predicts a text sample and quantifies the "surprise" encoded within language models (the lower the perplexity, the better the model). Despite its popularity, perplexity has been widely questioned as an unsatisfactory measure to compare the true merits of two different LLMs (Muhlgay et al., 2023), even for dense models although they significantly vary in model scale, training strategies, and design choices (encoder only, decoder only, *etc*.). To address this issue, several works (Li et al., 2023; Kaddour et al., 2023; Muhlgay et al., 2023; Zhang et al., 2023; Valmeekam et al., 2022; Liu et al., 2023a; Sawada et al., 2023; Qin et al., 2023; Zhuo, 2023; Lee et al., 2023) attempt to go beyond perplexity, and evaluate the capabilities of **dense LLMs** across commonsense reasoning, language understanding, reading comprehension, programming, *etc*. However, it is **critically important** to note that *all compressed models are derived from the same dense counterpart with high similarity* sharing exactly the same scale, training strategies, design choices, *etc*. Surprisingly, unlike dense LLMs, no such effort has been carried out to understand subtle changes in the capabilities of compressed LLMs with varying compression strength. Orthogonal to the recent trend to develop new compression algorithms, our work provides the **first** attempt to assess the true merits and limitations of existing SoTA LLM compression algorithms, to provide a fair and detailed playground to develop better compression algorithms. We focus on the *case of compressed LLMs* because we observe the profound failure of perplexity in capturing the delicate performance variations across varying LLM compressions.

Figure 1(Top) illustrates the change in perplexity of SoTA compression methods (pruning and quantization), such as SparseGPT, Wanda, GPTQ and baseline one-shot magnitude-based pruning on Vicuna-7B, 13B, and 33B (Chiang et al., 2023). Clearly, the perplexity (↓) of all models does not show any significant variation up to 45-60%, with a complete failure to capture subtle changes in the abilities of LLMs when compressed. It is also interesting to observe that to a certain degree of sparsity (∼ 30%), all SoTA pruning methods have almost similar performance as the simple baseline of

one-shot magnitude-based pruning, which raises questions about their true merits within this sparsity range. Figure 1(Bottom) show the response of Vicuna-7B model when compressed with Magnitude, SparseGPT, and Wanda by 50% and quantized up to 4-bit. The uncompressed Vicuna-7B was successfully able to generate the correct answer, but all compressed versions failed to respond correctly, hallucinating with either wrong facts or irrelevant responses.

## 3 LLM-KICK: UNVEILING TRUE MERITS OF LLM COMPRESSION

**LLM-KICK**, short for ***K**nowledge-**I**nstensive Compressed LLM Benchmar**K***, is crafted to bring the attention of LLM compression community towards incompetence of perplexity to correctly reflect subtle changes in the ability of LLMs derived from dense counterparts with varying compression strength. LLM-KICK consists of a suite of challenging, realistic, and diverse task settings of high practical importance and datasets that can empower a systematic understanding of how existing LLM compression strategies truly perform in preserving performance despite having similar perplexity. Our work thoroughly investigates proclaimed merits/limitations of pruned and quantized LLMs for language understanding, reasoning, generation, in-context retrieval, in-context summarization, *etc*.

Specifically, LLM-KICK consists of 3 broad task settings to study how compression impacts knowledge encoded during pre-training, how compressed LLMs perform tasks when required knowledge is augmented in-context, and how well compressed LLMs perform instruction following. To compartmentalize task difficulty and diversity, we include factoid-based QA, multiple-choice reasoning-based QA, in-context retrieval augmented QA, in-context text summarization, and instruction-based free-form text generation. Instead of creating new datasets, we carefully curate LLM-KICK from prior works and open-source GitHub repositories which have been widely accepted by researchers, but yet not explored by the LLM compression researchers. Our detailed prompt design strategies for different task settings can be found in Appendix A.2.

To reduce the expense of redundant experiments and clutter in results, our work primarily focuses on the top-2 existing training-free and data-free LLM pruning techniques (*i.e.*, SparseGPT (Frantar & Alistarh, 2023) and Wanda (Sun et al., 2023)), along with the baseline of One-shot Magnitude-based Pruning (Han et al., 2016), plus a popular quantization technique (GPTQ) among recently available choices (Lin et al., 2023a; Frantar et al., 2022; Dettmers et al., 2023c). We consider two types of sparsities: ($i$) **Unstructured Sparsity**: individual model weights are zeroed out independently, leading to irregular zero patterns (LeCun et al., 1990; Han et al., 2016); and ($ii$) **Structured N:M Sparsity**: a fine-grained sparsity pattern in which only *N* weights are non-zero for every continuous *M* weights (Nvidia, 2020; Zhou et al., 2021). We use Vicuna models for experiments, which are open-source chatbot models trained by fine-tuning LLaMA (Chiang et al., 2023) on user-shared conversations collected from ShareGPT, and have demonstrated impressive 90% quality of OpenAI ChatGPT and Google Bard. Note that the aim of this work is not limited to identifying the failure cases of SoTA pruning methods, but instead provides an in-depth lookup of LLM's ability under compression, and bring new insights which include highlighting observations that work in favor of current SoTA compression methods.

Formally, we study the performance drop of LLMs after compression (without fine-tuning) with respect to their dense counterparts using a compression algorithm $\mathtt{C}$. For a pre-trained LLM $f(x; \theta)$, a compressed LLM is a network $f_{\mathrm{comp}}(x; \theta_{\mathtt{C}})$, which is a copy of $f(x; \theta)$ with some weights fixed to 0 indicated by the pruning mask $m_{\mathtt{C}}$ in the case of pruning, or quantized to $k_{\mathtt{C}}$-bit using a quantization algorithm. Next, we define *matching* compressed LLM.

> **Matching Compressed LLM:** A compressed LLM $f_{\mathrm{comp}}(x; \theta_{\mathtt{C}})$ is *matching* for a compression algorithm $\mathtt{C}$ on task $\mathtt{T}$, if it results in performance no less than $\epsilon_0$ (compression tolerance regime) in comparison with $f(x; \theta, \mathtt{T})$. In this work, we consider $\epsilon_0$ to be $\leq 5\%$ of the performance of $f(x; \theta, \mathtt{T})$.

Note that $\epsilon_0$ is a simple indicator of the tolerance level of performance drop when we start compressing any LLM. Many prior works (Chen et al., 2020b; Jaiswal et al., 2023a) consider matching thresholds to be the same as the dense subnetwork performance or within the margins of 1%. However, in our work, we carefully relaxed it to 5% performance drop as an acceptable tolerance (before

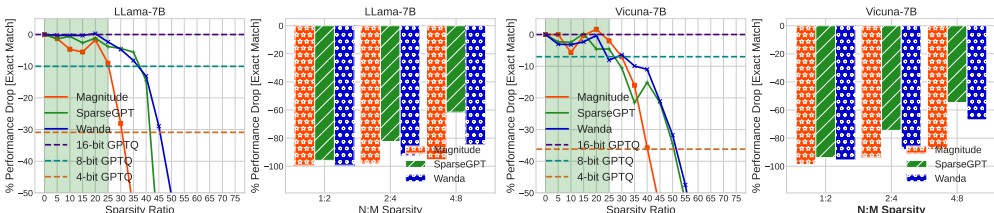

Figure 2: **Compressed LLMs for Factoid-based QA.** Performance comparison of compressed LLMs on Factoid-QA task using FreebaseQA (Jiang et al., 2019). Results (average across 3 independent runs) presented are for structured (N:M sparsity), unstructured sparsity, and quantization.

calling the compressed model useless) keeping in mind that the performance of compressed LLM on any of our task categories/disciplines remains above the random guess.

## 3.1 SETTING 1: HOW WELL COMPRESSED LLMS ACCESS REMAINING KNOWLEDGE?

### ① Factoid-based Question Answering

**Task Definition and Rationale.** Factoid-based Question Answering (Factoid-QA) (Iyyer et al., 2014), which asks precise facts about entities, is a long-standing problem in NLP. A typical Factoid-QA task aims to search for entities or entity attributes from a knowledge graph, and it is widely used as a tool in academia, commercial search engines, and conversational assistants. Modern LLMs are trained on gigantic text corpora ingesting a large amount of world knowledge about entities and their relationships during pre-training, and have unique abilities to generate factually correct responses to user queries. In this task setting, we aim to investigate *how compression impacts LLMs' ability to answer natural language questions using facts, i.e., entities or attributes knowledge ingested within them during pre-training.*

**Dataset Details.** We use FreebaseQA (Jiang et al., 2019) which is a dataset for open-domain QA over the Freebase knowledge graph. The QA pairs are collected from various sources, including the TriviaQA dataset (Joshi et al., 2017) and other trivia websites (QuizBalls, QuizZone, KnowQuiz), and are matched against Freebase to generate relevant subject-predicate-object triples that were further verified by human annotators. TriviaQA dataset shows rich linguistic variation and complexity, making it a good testbed for evaluating knowledge ingested within LLMs.

**Results and Analysis.** The results of various LLM compression methods are demonstrated in Figure 2. Our primary observations include: ① All SoTA LLM pruning methods **seemingly fail** to find matching sparse LLMs, *even at trivial sparsities such as 30-35%.* While several methods maintain the matching performance at 20-25% sparsity, their performance starts to drop significantly after that undergoing a *catastrophic failure* as sparsity ratio increases. This is in contrast with the claim made by SoTA pruning methods that pruning up to 50-60% of LLMs doesn't have any significant degradation on performance. ② All pruning methods **doesn't work** for fine-grained *structured N:M sparsity patterns with performance drop as severe as ≥50%.* ③ ~8-10% drop in performance for non-aggressive 8-bit quantization indicates that along with chasing for aggressive quantization levels (1-2 bits), it is also important to focus on yet unsolved 8-bit quantization.

### ② Multiple-Choice Reasoning based Question Answering

**Task Formulation and Rationale.** Multiple-Choice Reasoning based QA (MCR-QA) uses a natural prompting approach to present the question and answer options to the LLMs jointly, and have it output the symbol (*e.g.*, "A") associated with its chosen answer option. It allows the model to explicitly compare answer options. In this setting, we aim to investigate *compressed LLMs' ability to understand natural language questions, effectively reason using knowledge remaining within them, and successfully associate the correct answer among the given answer options with the symbols that represent them; potentially minimizing the effect of tokenization and exact answer generation.*

**Dataset Details.** We use the popular MMLU (Massive Multitask Language Understanding) benchmark which covers 50+ subjects across STEM, Humanities, Social Sciences, and more (Hendrycks et al., 2020). It ranges in difficulty from an elementary level to an advanced professional level, and it tests both world knowledge and problem-solving ability of LLMs. The granularity and breadth of subjects make it ideal for fine-grained evaluation of compressed LLMs' blind spots.

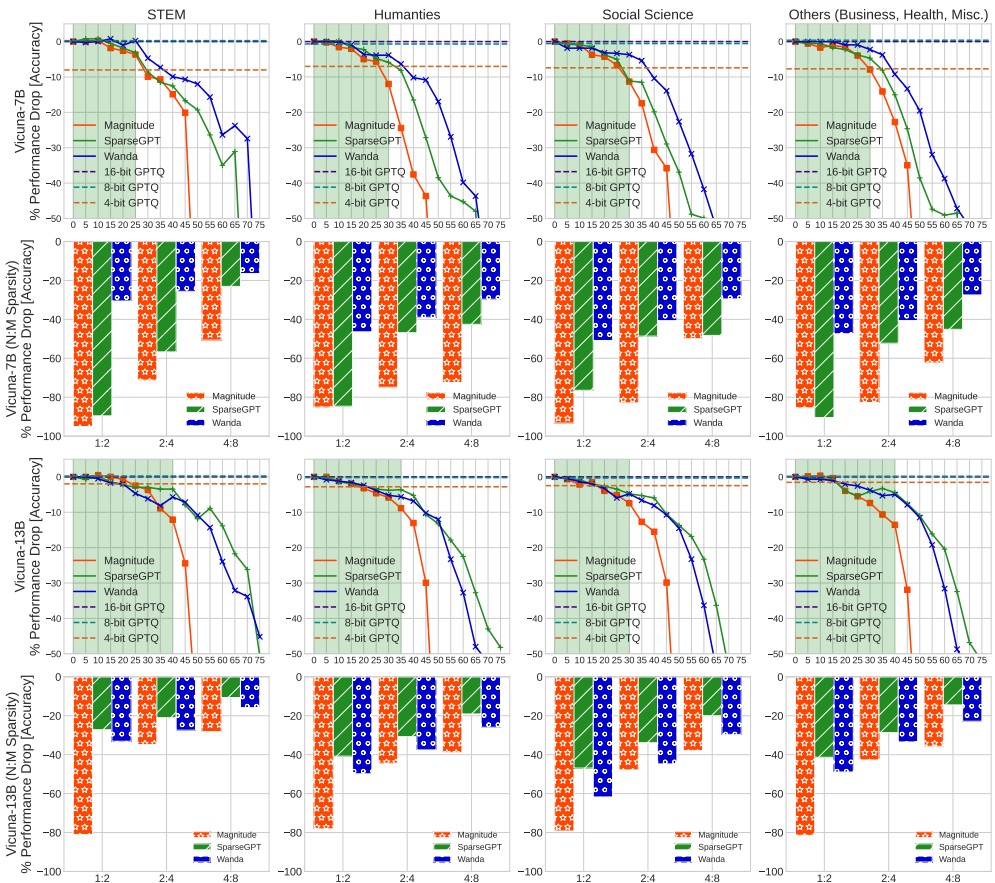

Figure 3: **Compressed LLMs for Multiple-Choice Reasoning based QA.** Performance comparison of compressed LLMs on MCR-QA tasks using the MMLU benchmark (Hendrycks et al., 2020). Results (average across 3 independent runs) presented are for structured (N:M sparsity), unstructured sparsity, and quantization.

**Results and Analysis.** The results of various LLM compression methods are demonstrated in Figure 3. Our primary observations include: ① Despite a similar matching compression regime (∼ 20-40%) to Factoid-QA, the abrupt performance drop of all SoTA pruning methods for MMLU is comparatively subtle due to relaxing the task setting from exact answer generation to correct answer selection. ② *No matching compressed LLMs* are found for N:M structured sparsity. ③ SoTA LLM quantization is **seemingly more successful** than SoTA pruning methods: we found 8-bit and 4-bit compressed LLM to be matching for Vicuna-7B and Vicuna-13B, respectively. ④ Interestingly, both quantization and pruning have comparatively higher performance drop for Humanities and Social Science wrt. STEM, which indicates **compression impacts some disciplines more than others**. ⑤ Surprisingly, within the compression tolerance regime, simple one-shot magnitude pruning seems to perform quite well in comparison with SoTA pruning method, illustrating its high effectiveness.

## 3.2   SETTING 2: HOW WELL COMPRESSED LLMs SYNTHESIZE AUGMENTED KNOWLEDGE?

### ① In-context Retrieval Augmented Question Answering

**Task Formulation and Rationale.** In-context Retrieval-Augmented Question Answering (ICRA-QA) (Ram et al., 2023) grounds the LLM answer generation by conditioning on relevant documents retrieved from an external knowledge source using retrieval algorithms like BM25. Our ICRA-QA evaluation system includes two high-level components: ⓐ *document selection*, selecting the set of documents upon which to condition; and ⓑ *document reading*, determining how to incorporate the selected documents into the LLM answer process, which requires extracting correct answer phrases from conditioned documents. To discount the impact of the lost encoded knowledge during compression, ICRA-QA **augments** the required relevant knowledge for QA task directly within the

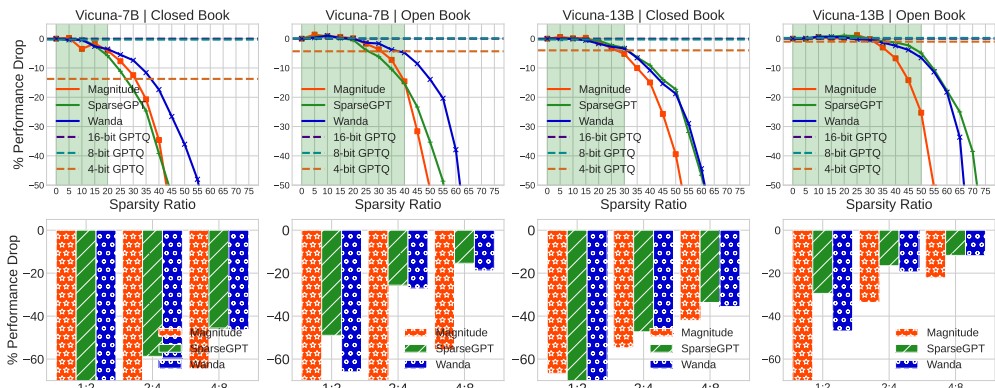

Figure 4: **Compressed LLMs for In-context Retrieval Augmented QA.** Performance comparison of compressed LLMs on ICRA-QA task. We present head-to-head comparison of closed-book evaluation (no external knowledge is augmented in-context) with open-book evaluation (external knowledge is augmented in-context). Results (average across 3 independent runs) presented are for structured N:M sparsity, unstructured sparsity, and quantization.

prompt context. In this task setting, we aim to evaluate *compressed LLMs' ability to synthesize long in-context knowledge provided within input prompts, and locate and retrieve correct answers within it.* We also present a head-to-head comparison of how augmented knowledge can work as a *remedy* to supplement the lost knowledge under compression.

**Dataset Details.** We use TriviaQA (Joshi et al., 2017) for evaluation, a popular reading comprehension dataset which includes 95K question-answer pairs authored by trivia enthusiasts and independently gathered evidence documents, six per question on average, that provide high-quality distant supervision for answering the questions.

**Results and Analysis.** The results of various LLM compression methods are demonstrated in Figure 17. The closed-book setting differs from ICRA-QA (*i.e.*, using the open-book setting) only in terms of whether conditioning on relevant documents retrieved from an external knowledge source. Our key findings are: ① When compressed LLMs are conditioned on external knowledge (open book) and assigned the task of in-context retrievers, *i.e.*, extracting correct answer phrases from in-context knowledge, they *perform significantly well* even in extremely high compression regime. Vicuna-7B can remain matching till ∼40% sparsity and 8-bit quantization, while Vicuna-13B can remain matching up to ∼50% sparsity and 4-bit quantization. Our experimental results send a positive signal that even if *high compression leads to significant knowledge loss, it doesn't leave LLMs completely useless*, and they still work as robust in-context retrievers. ② Despite we observe a significant benefit while conditioning external knowledge, **no** matching compressed LLM can be identified for N:M sparsity. ③ Again, we observe surprisingly good performance of simple one-shot unstructured magnitude pruning wrt. SparseGPT (second-order pruning) and Wanda (activation-based pruning) that rely on calibration data.

② **In-Context Text Summarization**

**Task Formulation and Details.** Modern LLMs have shown astonishing success in summarizing long-context documents in both abstractive and extractive settings. However, it is **yet not explored** how compression impacts LLMs' capability for summarization. In this task setting, we aim to investigate *compressed LLMs' ability to hold onto consistency, coherence, fluency, and relevance when prompted to summarize textual information of varying length (small, medium, and large) in abstractive setting (Jain et al., 2023).* For evaluation, similar to Zheng et al. (2023), we propose to use GPT-4 as a judge, which compares the compressed LLM generated summaries wrt. GPT-3.5 (text-davinci-003) generated summaries. Detailed evaluation settings can be found in Appendix A.3.

**Dataset Details.** We use a popular summarization dataset CNN/DailyMail (Chen et al., 2016) for evaluation, which is an English-language dataset containing just over 300k unique news articles written by journalists at CNN and DailyMail. We created 3 subset categories {small (≤470 words), medium (≥470 and ≤ 790 words), and large (≥ 790 words)} of stories, each with 100 articles reflecting word distribution of CNN/DailyMail to minimize OpenAI API costs.

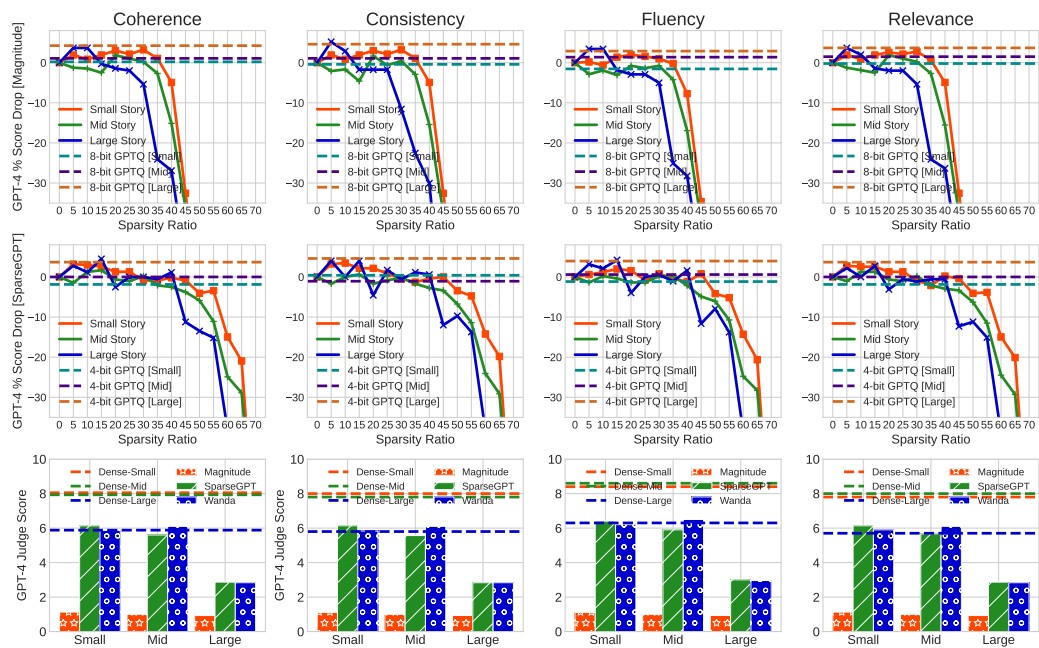

Figure 5: **Compressed LLMs for In-Context Summarization.** Performance comparison of compressed Vicuna-7B for in-context summarization of small, medium, and large stories while preserving coherence, consistency, fluency, and relevance. Results (average across 3 independent runs) presented are for structured (2:4 sparsity - Row 3), unstructured sparsity, and quantization.

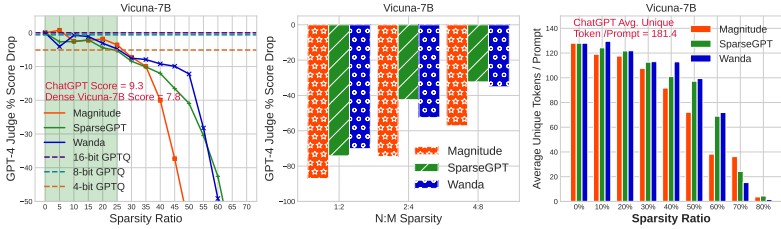

Figure 6: **Compressed LLMs for Instruction Following.** LLM-as-a-Judge: GPT-4 based evaluation of compressed Vicuna-7B response wrt. ChatGPT (`davici-003`). (Left) unstructured sparsity; (middle) structured N:M sparsity; (c) comparison of average unique token counts generated by compressed Vicuna-7B for 80 prompts across 10 different categories.

**Results and Analysis.** Results are summarized in Figure 5. We summarize our main observations as: ① All pruning and quantization methods tend to perform **surprisingly well** for in-context summarization, preserving high consistency, coherence, fluency, and relevance in generated summaries, which is an *encouraging observation in favor compression*. ② With increasing context length (*i.e.*, long stories), we observe a *sharper performance drop* for compressed LLMs, which highlights that compression impacts LLMs' ability to synthesize and summarize longer context lengths. ③ Quantization again seems to perform better than SoTA pruning methods, and surprisingly benefiting positively over the dense model performance. ④ No matching compressed LLM can be identified for 2:4 structured sparsity.

## 3.3 SETTING 3: HOW WELL COMPRESSED LLMs PERFORM INSTRUCTION FOLLOWING?

**Task Formulation and Rationale.** In this task setting, we investigate *compressed LLMs' ability to answer open-ended questions and evaluate their multi-turn conversational and instruction-following ability – two critical elements for human preference*. Evaluating AI chatbots is a challenging task, as it requires examining language understanding, reasoning, and context awareness. To compare the performance of compressed LLMs' responses, we closely follow the prompt design setting in MT-Bench (Zheng et al., 2023) using GPT-4 as a judge. We prompt GPT-4 to rate the answers gen-

erated by compressed LLMs wrt. GPT-3.5 (text-davinci-003) model based on varying metrics (*e.g.*, correctness, helpfulness, logic, accuracy, *etc.*) on a scale of `[0-10]` with detailed explanations.

**Dataset Details.** We rely on the 80 high quality multi-turn questions identified in MT-Bench (Zheng et al., 2023). This setting covers common-use human-centric interaction with LLMs, and focuses on challenging questions to differentiate models. We used 8 common categories of user prompts to guide the prompt construction to interact with compressed LLMs: writing, roleplay, extraction, reasoning, math, coding, *etc*. For each category, we adopted manually designed 10 multi-turn questions from MT-Bench to evaluate our compressed models. Details can be found in Appendix A.4.

**Results and Analysis.** Results are summarized in Figure 6. Our primary observations are: ① Unlike in-context text summarization, in this task setting, compressed LLMs have to access the knowledge to respond to conversations maintaining high helpfulness, relevance, accuracy, and detail. We again observe that compressed LLMs with various pruning methods are *matching only up to sparsity ratio of ∼ 25%*. ② Surprisingly, in the matching regime, the simple baseline of one-shot magnitude pruning performs *comparable or slightly better* than SoTA pruning methods. ③ *No matching* subnetwork can be identified for N:M sparsity. ④ Interestingly, our average generated unique token analysis in Figure 6(c) illustrates that compressed LLMs lose the ability to generate distinct unique content, instead, they can only produce more repetitive texts.

## 4 ADDITIONAL RESULTS AND DISCUSSIONS

**Small-Dense vs. Large-Sparse: which is favorable?** We attempt to understand an interesting question: *if pruned LLMs with larger architecture (Large-Sparse) is better than smaller dense models with similar parameter count (Small-Dense)?* Pruning large LLMs doesn't come for free, and it is important to investigate if the cost of pruning can be reflected in the performance benefit of Large-Sparse models. To our surprise, in comparison with dense Vicuna-7B (MMLU accuracy 46.7%), we found compressed Vicuna-13B with exactly similar parameter count (46.16% sparsity) of 7 billion using one-shot magnitude, Wanda, SparseGPT can only achieve MMLU accuracy of 31.7%, 45.3%, and 46.3%, respectively. This is a clear indication that current sparsity algorithms are **not yet** up to a stage where the cost of pruning can be justified by performance benefits obtained from large-sparse compressed models.

**How many calibration data samples are needed?** We attempt to analyze how calibration dependent pruning methods (Wanda and SparseGPT) perform with varying amount of calibration samples. Figure 7 illustrates the zero-shot performance of 50% & 70% pruned Vicuna-7B using Wanda and SparseGPT on knowledge-intensive MMLU benchmark. It is interesting to observe that calibration sample count plays a vital role in preserving the performance of SparseGPT unlike Wanda. Note that at high sparsity ratio (70%), Wanda cannot recover any performance; SparseGPT surprisingly benefits noticeably from calibration. This suggests that *carefully selected calibration samples can play a vital role* in designing better pruning algorithms to compress LLMs even up to significantly high sparsity.

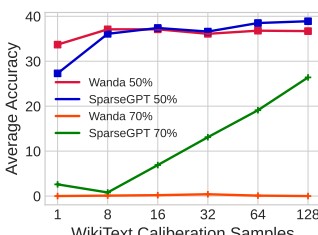

Figure 7: Zero-shot performance of 50% & 70% pruned Vicuna-7B wrt. calibration sample counts.

## 5 CONCLUSION AND LIMITATIONS

In this paper, we propose to explore the effectiveness of SoTA compression methods beyond perplexity to address the inability of perplexity to capture the subtle variations incurred during the derivation of compressed LLMs from their dense counterparts. Our work introduces **K**nowledge-**I**ntensive **C**ompressed LLM Benchmar**K** (LLM-KICK) to facilitate a fair and holistic evaluation by unveiling many merits and pitfalls of SoTA compression methods. Our study reveals that compression significantly impacts the knowledge encoded in LLMs during pre-training, compressed LLMs perform quite well with knowledge augmented in-context settings. We primarily restrict our evaluation to Vicuna (decoder-only architecture) due to its open-source license, high performance, and instruction-following ability. For future work, we aim to investigate how the lost knowledge due to compression can be recovered using parameter-efficient fine-tuning methods, *e.g.*, LoRA (Hu et al., 2021) and QLoRA (Dettmers et al., 2023b).

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

## A    APPENDIX

### A.1    RELATED WORKS

#### A.1.1    SPARSITY IN LARGE LANGUAGE MODELS

The advent of large-scale pre-trained models has led to the development of advanced post-training pruning methods, aiming to enhance the cost-effectiveness of these expansive models (Sanh et al., 2020; Chen et al., 2020a; Jaiswal et al., 2023b; Zafrir et al., 2021; Kurtic et al., 2022; Xu et al., 2021; Lagunas et al., 2021; Zhang et al., 2022; Frantar et al., 2021; Jaiswal et al., 2023a; Ma et al., 2023; Ji et al., 2023). Among them, Frantar et al. (2021) extend second-order pruning to the BERT-level scale, enabling the pruning of blocks of weights and achieving state-of-the-art results for sparse BERT. Frantar & Alistarh (2023) introduce SparseGPT for pruning large language models (LLMs) in a single shot without requiring re-training or fine-tuning. They leverage column-wise second-order pruning, and successfully remove 100B weights from OPT-175B without a significant increase in perplexity. More recently, Sun et al. (2023) propose a straightforward pruning method that takes both weights and activations into account, demonstrating comparable performance to Frantar & Alistarh (2023). Li et al. (2022) reveal that activation sparsity is a prevalent phenomenon in Transformers (90% of intermediate output), yielding another opportunity for acceleration. Liu et al. (2023b) introduce a large-scale SMC-Bench, indicating that state-of-the-art magnitude- and/or gradient-based sparse algorithms fall short when applied out-of-the-box to larger-scale models and a selected of complex downstream tasks.

#### A.1.2    QUANTIZATION IN LARGE LANGUAGE MODELS

With the recent open-source releases of language models like BLOOM, Vicuna, LLaMa, OPT, etc., quantization has emerged as a widely embraced technique to alleviate the storage and computational overhead of deep learning models. Recent research endeavors have harnessed quantization to compress LLMs and they can be classified into the two mentioned approaches: Quantization-Aware Training (QAT), and Post-Training Quantization (PTQ). In QAT, the quantization objective is embedded into the LLM training process, enabling them to adapt to low-precision representations and handle precision loss caused by quantization. LLM-QAT (Liu et al., 2023c) proposes a data-free distillation method that leverages generations produced by the pre-trained model, preserving the original output distribution and allows quantizing LLaMa models independent of its training data. PEQA (Kim et al., 2023) operates through a dual-stage process: initially, the parameter matrix of each fully-connected layer undergoes quantization into a matrix of low-bit integers and a scalar vector; subsequently, fine-tuning occurs on the scalar vector for each downstream task. QLoRA (Dettmers et al., 2023a) proposes an efficient finetuning approach that reduces memory usage enough to finetune a 65B parameter model on a single 48GB GPU while preserving full 16-bit finetuning task performance by backpropagating gradients through a frozen, 4-bit quantized pre-trained language model into Low Rank Adapters (LoRA). PTQ involves quantizing the parameters of LLMs after the completion of the LLM's training phase. GPTQ (Frantar et al., 2022) proposes a novel layer-wise quantization technique based on approximate second-order information resulting a bitwidth reduction to 3 or 4 bits per weight, with minimal accuracy loss compared to the uncompressed version. AWQ (Lin et al., 2023a) based on the observation that weights are not equally important: protecting only 1% of salient weights can greatly reduce quantization error, employs an activation-aware approach by considering the significance of weight channels corresponding to larger activation magnitudes. SpQR (Dettmers et al., 2023c) works by identifying and isolating outlier weights, which cause particularly-large quantization errors, and storing them in higher precision, while compressing all other weights to 3-4 bits, and achieves relative accuracy losses of less than 1% in perplexity for highly-accurate LLaMA and Falcon LLMs.

#### A.1.3    LARGE LANGUAGE MODELS AND EVALUATION

Large language models (LLMs) are gaining increasing popularity in both academia and industry playing vital role in both research and daily use. With increasing popularity, several works (Li et al., 2023; Kaddour et al., 2023; Muhlgay et al., 2023; Zhang et al., 2023; Valmeekam et al., 2022; Liu et al., 2023a; Sawada et al., 2023; Qin et al., 2023; Zhuo, 2023; Lee et al., 2023) attempt to go beyond conventional perplexity to evaluate performance of LLMs across factuality, commonsense

reasoning, language understanding, reading comprehension, programming, instruction following abilities, *etc*. Muhlgay et al. (2023) propose a new metric FACTOR to understand factuality correct information in the LLM generated text. It found that although FACTOR accuracy and LMM perplexity tend to be highly correlated but sometimes induce different orderings between LMMs. They reported that pairs of models can share similar perplexity but differ significantly in terms of FACTOR accuracy. Lee et al. (2023) evaluate the performance and alignment of LLM distribution with humans using two different techniques: Monte Carlo Reconstruction (MCR) and Log Probability Reconstruction (LPR); and found LLMs exhibit limited ability in solving NLI tasks and simultaneously fail to capture human disagreement distribution. Zhang et al. (2023) attempt to investigate promise for automatic summarization with respect to human summary writers and found that LMM summaries are judged to be on par with human written summaries. Valmeekam et al. (2022) propose an extensible assessment framework to test the capabilities of LLMs on reasoning about actions and change, a central aspect of human intelligence and found that GPT-3 and BLOOM have dismal performance on these benchmarks. Despite these efforts to investigate the performance of dense LLMs comprehensively, it is surprising that no such efforts have been yet carried out for a more daunting case of compressed LLMs, which are derived from dense counterparts sharing significantly high similarity with them. Our work is first attempt to address this gap and encourage sparse community researchers to go beyond perplexity to evaluate the true merits and drawbacks of compression methods.

## A.2 PROMPT DESIGN AND EXAMPLES FOR DIFFERENT TASK SETTINGS IN LLM-KICK

### A.2.1 FACTOID-BASED QA

**Prompt Design:** Please give answer to this question: `<QUESTION>` The answer is

**Example:** Please give answer to this question: `The film '10 things I hate about you' is based on which Shakespeare play?` The answer is

**Model Response:** Please give answer to this question: `The film '10 things I hate about you' is based on which Shakespeare play?` The answer is `the taming of the shrew`.

### A.2.2 MULTIPLE-CHOICE REASONING-BASED QA

**Prompt Design:** The following are multiple choice questions (with answers) about `<SUBJECT NAME>`.\n\n`<QUESTION>` \nA. `<OPTION 1>`\nB. `<OPTION 2>`\nC. `<OPTION 3>`\nD. `<OPTION 4>`\n Answer:

**Example:** The following are multiple choice questions (with answers) about `algebra.`\n\n `Find the degree for the given field extension Q(sqrt(2), sqrt(3), sqrt(18)) over Q.` \nA. 0\nB. 4\nC. 2\nD.6\n Answer:

**Model Response:** The following are multiple choice questions (with answers) about `algebra.`\n\n `Find the degree for the given field extension Q(sqrt(2), sqrt(3), sqrt(18)) over Q.` \nA. 0\nB. 4\nC. 2\nD.6\n Answer: `B`

### A.2.3 IN-CONTEXT RETRIEVAL AUGMENTED QUESTION ANSWERING

① **Closed Book Setting:** For closed-book setting, we adopted the prompt from Touvron et al. (2023) as follows.

> **Prompt Design:**  Answer these questions:\n\nQ: <QUESTION>\n A:
>
> **Example:**  Answer these questions:\n\nQ: `Who was the man behind The Chipmunks?\n A:`
>
> **Model Response:**  Answer these questions:\n\nQ: `Who was the man behind The Chipmunks?\n A:` `The man behind The Chipmunks was David Sarge, who was the founder of the Alphaville Virtual Real Estate Company.`

② **Open Book Setting:** For open-book setting, we extend the above prompt as follows.

> **Prompt Design:**  <EVIDENCE>\n Answer these questions:\nQ: <QUESTION>\n A:
>
> **Example:**  `''Alvin and the Chipmunks (2007) — IMDb IMDb 17 January 2017 4:34 PM, UTC NEWS. A struggling songwriter named Dave Seville finds success ..."\n` Answer these questions:\n Q: `Who was the man behind The Chipmunks?\n` A:
>
> **Model Response:**  `''Alvin and the Chipmunks (2007) — IMDb IMDb 17 January 2017 4:34 PM, UTC NEWS. A struggling songwriter named Dave Seville finds success ..."\n` Answer these questions:\n Q: `Who was the man behind The Chipmunks?\n` A: `Dave Seville.`

### A.2.4  IN-CONTEXT TEXT SUMMARIZATION

> **Prompt Design:**  A chat between a curious user and an artificial intelligence assistant. The assistant gives helpful, detailed, and polite answers to the user's questions. USER: `Summarize the given story in less than 150 words while preserving high coherence, consistency, fluency, and relevance.\n\n` <STORY>. ASSISTANT:
>
> **Example:**  A chat between a curious user and an artificial intelligence assistant. The assistant gives helpful, detailed, and polite answers to the user's questions. USER: `Summarize the given story in less than 150 words while preserving high coherence, consistency, fluency, and relevance.\n\nLibyan and U.S. officials say the two governments held face-to-face talks in Tunisia ...have denied previous reports of talks with the government.` ASSISTANT:

**Model Response:**  The model response of one-shot magnitude pruned Vicuna-7B ASSISTANT is shown in Figure 8.

Libyan and U.S. officials held face-to-face talks in Tunisia over the weekend, but Washington's sole point of the meeting was to repeat its demand that Libyan leader Moammar Gadhafi "must go". Libyan government spokesman Musa Ibrahim called the talks "a first step" and stated that the two sides were ready to discuss ideas to end the conflict and repair the damaged relationship between Libya and the U.S. However, a U.S. official familiar with the meeting said the purpose was "for us to convey directly that Gadhafi must go". NATO has been bombing Libya under a U.N. mandate to protect civilians from Gadhafi's forces, who are battling a rebellion that has claimed control of the eastern half of the country. Despite previous reports of talks between the rebels and the government, Libyan rebels have denied any such talks.

Figure 8: Output response of 10% compressed (unstructured one-shot) Vicuna-7b ASSISTANT.

### A.2.5 MULTI-TURN CONVERSATION AND INSTRUCTION FOLLOWING

> **Prompt Design:** A chat between a curious user and an artificial intelligence assistant. The assistant gives helpful, detailed, and polite answers to the user's questions. USER: `<QUESTION>` ASSISTANT:
>
> **Example:** A chat between a curious user and an artificial intelligence assistant. The assistant gives helpful, detailed, and polite answers to the user's questions. USER: `How can I improve my time management skills?` ASSISTANT:

**Model Response:** The model response of one-shot magnitude pruned Vicuna-7B ASSISTANT is shown in Figure 9.

Improving your time management skills can help you be more productive and efficient in your daily tasks and activities. Here are some tips that may be helpful:

1. Set clear goals: Identify your priorities and set clear, specific goals for what you want to accomplish. This will help you focus your efforts and use your time effectively.
2. Prioritize your tasks: Make a to-do list and prioritize your tasks based on their importance and urgency. This can help you allocate your time more efficiently and avoid procrastination.
3. Estimate your time: Before starting a task, estimate how long it will take you to complete it. This can help you plan your time more effectively and avoid getting overwhelmed by your workload.
4. Use time management tools: There are many time management tools available, such as calendars, schedulers, and reminders. Use these tools to help you keep track of your tasks and deadlines.
5. Learn to say no: Be mindful of your time and learn to say no to activities that are not essential or that you don't have the time to commit to.
6. Take breaks: Taking regular breaks can help you stay focused and productive. Set a timer for your work sessions and take a break after the timer goes off.
7. Eliminate distractions: Identify and eliminate distractions, such as social media, email, or television, that can take up your time and prevent you from focusing on your tasks.

Remember, time management is a skill that takes practice, so be patient with yourself and keep working at it.

Figure 9: Output response of 10% compressed (unstructured one-shot) Vicuna-7b ASSISTANT.

### A.3 IN-CONTEXT SUMMARIZATION EVALUATION SETTINGS

For evaluating the performance of LLMs to generate high-quality in-context summarization, we focus on consistency, coherence, fluency, and relevance metrics. We prompt GPT-4 which has been recently identified to be highly effective as an automated evaluation framework for benchmark generation and performance assessments, to evaluate these metrics in comparison to the summaries generated by GPT-3.5. Examples of our prompts used for evaluating with GPT-4 Judge are shown in Figure 10. We also provide an example of GPT-4 Judge output in Figure 11.

**IN-CONTEXT SUMMARIZATION EVALUATION PROMPT >>** "You are a helpful and precise assistant for checking the quality of the summarization of two stories within 150 words.", "prompt_template": "[STORY]\n{story}\n\n[The Start of Assistant 1's Summary]\n{summary_1}\n\n[The End of Assistant 1's Summary]\n\n[The Start of Assistant 2's Summary]\n{summary_2}\n\n[The End of Assistant 2's Summary]\n\n[System]\n{prompt}\n\n", "defaults": {"prompt": "We would like to request your feedback on the performance of two AI assistants in response to the user requested summary above.\nPlease rate the coherence, consistency, fluency, and relevance of summary generated. Each assistant receives a score on a scale of 1 to 10 for coherence, consistency, fluency and relevance, where a higher score indicates better overall performance.\nPlease first output four lines containing only two values indicating the scores for Assistant 1 and 2, respectively for each four metrices. The two scores are separated by a space. In the subsequent line, please provide a comprehensive explanation of your evaluation, avoiding any potential bias and ensuring that the order in which the responses were presented does not affect your judgment."}

Figure 10: Example of prompt used to evaluate the compressed LLM ASSISTANT *wrt.* GPT-3.5 ASSISTANT using GPT-4 as Judge on consistency, coherence, fluency, and relevance of generated summaries.

8 7
8 7
9 8
8 7
Assistant 1 provided a more comprehensive summary of the story, including more details about the political dynamics, the role of Michelle Obama, and the context of the Senate race in Iowa. The summary was coherent, consistent, and fluent, with all the key points from the story accurately represented. The relevance of the summary to the original story was high, as it captured the main points and provided a clear overview of the situation.

Assistant 2's summary was also coherent, consistent, and fluent, but it lacked some of the details that were included in Assistant 1's summary, such as the fact that Michelle Obama had already campaigned for another candidate earlier in the week, and the specific accusations being made by Braley against Ernst. The relevance of the summary to the original story was slightly lower than Assistant 1's, as it left out some key points. However, it still provided a good overview of the main points of the story.

Figure 11: GPT-4 Judge Evaluation of responses generated by GPT-3 (ASSISTANT 1) *wrt.* 10% compressed (unstructured one-shot) Vicuna-7b (ASSISTANT 2).

### A.4 INSTRUCTION FOLLOWING ABILITY EVALUATION SETTING

For evaluating the responses generated by compressed LLMs, we closely follow the prompt design settings of MT-Bench (Zheng et al., 2023) using GPT-4 as judge. We prompt GPT-4 to rate the answers generated by compressed LLMs wrt. GPT-3.5 (text-davinci-003) model based on varying metrics (eg. correctness, helpfulness, logic, accuracy, *etc.*) on a scale of `[0-10]` and provides a

detailed explanation behind the score. Examples of our prompts used during evaluation for questions as well as GPT-4 Judge response are as shown in Figure 12, and 13, respectively.

**GENERAL QUESTION PROMPT >>** You are a helpful and precise assistant for checking the quality of the answer.", "prompt_template": "
[Question]\n{question}\n\n[The Start of Assistant 1's Answer]\n{answer_1}\n\n[The End of Assistant 1's Answer]\n\n[The Start of Assistant 2's Answer]\n{answer_2}\n\n[The End of Assistant 2's Answer]\n\n[System]\n{prompt}\n\n", "defaults": {"prompt": "We would like to request your feedback on the performance of two AI assistants in response to the user question displayed above.\nPlease rate the helpfulness, relevance, accuracy, level of details, factual information, and length of their responses. Each assistant receives an overall score on a scale of 1 to 10, where a higher score indicates better overall performance.\nPlease first output a single line containing only two values indicating the scores for Assistant 1 and 2, respectively. The two scores are separated by a space. In the subsequent line, please provide a comprehensive explanation of your evaluation, avoiding any potential bias and ensuring that the order in which the responses were presented does not affect your judgment."}

**CODING QUESTION PROMPT >>** You are a helpful and precise assistant for checking the quality of the answer.", "prompt_template": "[Question]\n{question}\n\n[The Start of Assistant 1's Answer]\n{answer_1}\n\n[The End of Assistant 1's Answer]\n\n[The Start of Assistant 2's Answer]\n{answer_2}\n\n[The End of Assistant 2's Answer]\n\n[System]\n{prompt}\n\n", "defaults": {"prompt": "Your task is to evaluate the coding abilities of the above two assistants. They have been asked to implement a program to solve a given problem. Please review their code submissions, paying close attention to their problem-solving approach, code structure, readability, and the inclusion of helpful comments.\n\nPlease ensure that the assistants' submissions:\n\n1. Correctly implement the given problem statement.\n2. Contain accurate and efficient code.\n3. Include clear and concise comments that explain the code's logic and functionality.\n4. Adhere to proper coding standards and best practices.\n\nOnce you have carefully reviewed both submissions, provide detailed feedback on their strengths and weaknesses, along with any suggestions for improvement. You should first output a single line containing two scores on the scale of 1-10 (1: no code/no sense; 10: perfect) for Assistant 1 and 2, respectively. Then give extra comments starting from the next line."}

**MATHS QUESTION PROMPT >>** You are a helpful and precise assistant for checking the quality of the answer.", "prompt_template": "[Question]\n{question}\n\n[The Start of Assistant 1's Answer]\n{answer_1}\n\n[The End of Assistant 1's Answer]\n\n[The Start of Assistant 2's Answer]\n{answer_2}\n\n[The End of Assistant 2's Answer]\n\n[System]\n{prompt}\n\n", "defaults": {"prompt": "We would like to request your feedback on the mathematical proficiency of two AI assistants regarding the given user question displayed above.\nFirst, please solve the problem independently, without referring to the answers provided by Assistant 1 and Assistant 2.\nAfterward, please examine the problem-solving process of Assistant 1 and Assistant 2 step-by-step to ensure their correctness, identifying any incorrect steps if present. Your evaluation should take into account not only the answer but also the problem-solving steps.\nFinally, please output a Python tuple containing two numerical scores for Assistant 1 and Assistant 2, ranging from 1 to 10, respectively. If applicable, explain the reasons for any variations in their scores and determine which assistant performed better."}

Figure 12: Examples of prompts used for different categories to evaluate the compressed LLM ASSISTANT *wrt.* GPT-3.5 ASSISTANT using GPT-4 as a Judge.

9 8

Both assistants provided comprehensive and accurate responses to the question. They both identified key challenges such as access to quality education, teacher quality, the role of technology, and the importance of equity and inclusion.

Assistant 1's response was slightly more detailed and organized, providing a clear enumeration of the challenges and a brief explanation for each. The assistant also touched on the importance of addressing global issues in education, which is a crucial aspect of modern education. The conclusion about the need for collaboration among stakeholders was a thoughtful addition.

Assistant 2 also provided a good response, but it was slightly less detailed and organized compared to Assistant 1. The points were relevant and accurate, but the explanations were not as comprehensive. The assistant did, however, bring up the challenge of student engagement, which is a significant issue in education today. The point about globalization of education was also a unique and relevant addition.

Overall, both assistants did a good job, but Assistant 1's response was slightly more detailed and well-structured.

Figure 13: GPT4-as-a-Judge evaluation of responses generated by GPT-3 (ASSISTANT 1) *wrt.* 10% compressed (unstructured one-shot) Vicuna-7b (ASSISTANT 2).

## A.5 Useful Links for LLM-KICK

Table 1: Dataset and code link used in our work.

| Method / Dataset | Download URL |
|---|---|
| FreebaseQA (Jiang et al., 2019) | https://huggingface.co/datasets/freebase_qa |
| MMLU Benchmark (Hendrycks et al., 2020) | https://huggingface.co/datasets/freebase_qa |
| TriviaQA (Joshi et al., 2017) | https://huggingface.co/datasets/trivia_qa |
| MT-Bench (Zheng et al., 2023) | https://huggingface.co/datasets/HuggingFaceH4/mt_bench_prompts |
| CNN/DailyMail Summarization (Nallapati et al., 2016) | https://cs.nyu.edu/~kcho/DMQA/ |
| WikiText (Merity et al., 2016) | https://huggingface.co/datasets/HuggingFaceH4/mt_bench_prompts |
| Wanda (Sun et al., 2023) | https://github.com/locuslab/wanda |
| SparseGPT (Frantar & Alistarh, 2023) | https://github.com/IST-DASLab/sparsegpt |
| LLM-Judge (Zheng et al., 2023) | https://github.com/lm-sys/FastChat/tree/main/fastchat/llm_judge |
| GPTQ (Frantar et al., 2022) | https://github.com/qwopqwop200/GPTQ-for-LLaMa |

## A.6 Comparison with AWQ and LLM-int8

In this section, we considered evaluating AWQ (Lin et al., 2023b) and LLM.int8() (Dettmers et al., 2022) across our different task settings and we summarize our results on Vicuna-7B as in the following table. We observe that LLM.int8() despite its simplicity and ease-of-use, achieves better results than AWQ (8-bit), and GPTQ (8-bit) across all listed tasks.

| Task | GPTQ | AWQ | LLM-int8() |
|------|------|-----|------------|
| Factoid-QA | 60.14% | 60.31% | 61.02% |
| MCR-QA (MMLU) | 47.10% | 47.18% | 47.82% |
| Retrieval Augmented QA | 75.55% | 75.89% | 75.91% |
| Instruction Following (GPT4-Score) | 9.74 | 9.72 | 9.81 |

Table 2: Performance comparison of AWQ and LLM-int8() on LLM-KICK.

## A.7 UNDERSTANDING THE IMPACT OF K-SHOT FOR COMPRESSED LLMs

In this section, we aim to investigate how few-shot in-context learning examples can benefit SoTA pruning methods to preserve performance across various sparsity levels. Figure 14 illustrates the performance comparison of Vicuna-7B at varying sparsity ratios when augmented with k-shot in-context examples on MMLU benchmark. It is interesting to observe that k-shot in-context learning examples have **marginal impact** on dense network performance, while they **significantly help** in preserving the performance at high sparsity. Moreover, we found 2-3 examples are sufficient to retain the performance, and supplementing additional examples doesn't necessarily provide further noticeable benefits.

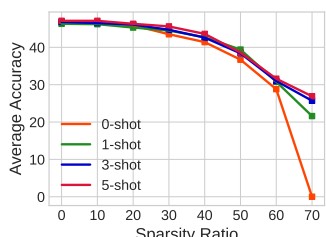

Figure 14: k-shot results of Vicuna-7B pruned with Wanda.

## A.8 SUMMARY OF VARIOUS PRUNING METHODS ON LLM-KICK

| Task | Pruning Method | 0% | 10% | 20% | 30% | 40% | 50% |
|------|----------------|-----|-----|-----|-----|-----|-----|
| Factoid-QA | Magnitude | 65.44 | 61.74 | 66.53 | 60.84 | 42.06 | 13.99 |
| | SparseGPT | 65.44 | 63.84 | 62.44 | 58.54 | 55.54 | 42.86 |
| | Wanda | 65.44 | 63.34 | 65.23 | 61.24 | 58.24 | 44.66 |
| MCR-QA (MMLU) | Magnitude | 0.471 | 0.466 | 0.455 | 0.422 | 0.339 | 0.050 |
| | SparseGPT | 0.471 | 0.470 | 0.460 | 0.437 | 0.395 | 0.308 |
| | Wanda | 0.471 | 0.469 | 0.460 | 0.455 | 0.425 | 0.386 |
| In-context Retrieval (Long Story: Coherence) | Magnitude | 5.883 | 6.112 | 5.855 | 5.567 | 4.329 | 1.233 |
| | SparseGPT | 5.883 | 6.033 | 5.533 | 6.067 | 5.567 | 5.067 |
| | Wanda | 5.883 | 6.0 | 5.783 | 5.933 | 5.267 | 5.033 |
| Instruction Following (GPT-4 Score) | Magnitude | 7.763 | 7.567 | 7.621 | 7.201 | 6.208 | 3.308 |
| | SparseGPT | 7.763 | 7.645 | 7.50 | 7.188 | 6.905 | 6.206 |
| | Wanda | 7.763 | 7.731 | 7.546 | 7.202 | 7.071 | 6.838 |

Table 3: Performance comparison of various pruning methods on Vicuna-7B with LLM-KICK.

## A.9 ADDITIONAL RESULTS ON LLaMa-2

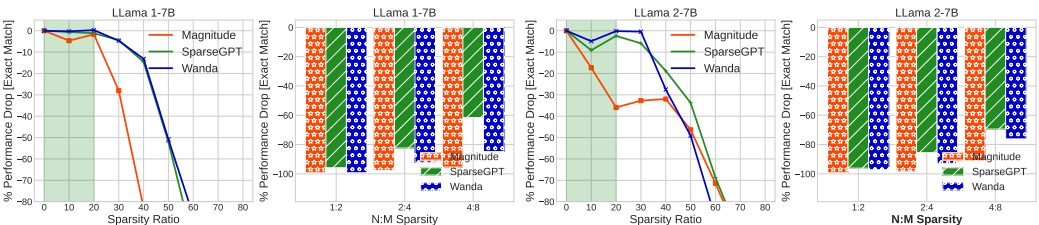

Figure 15: **Compressed LLMs for Factoid-based QA.** Performance comparison of compressed LLMs (LLaMa 1 & 2) on Factoid-QA task using FreebaseQA (Jiang et al., 2019). Results presented are for structured (N:M sparsity) and unstructured sparsity.

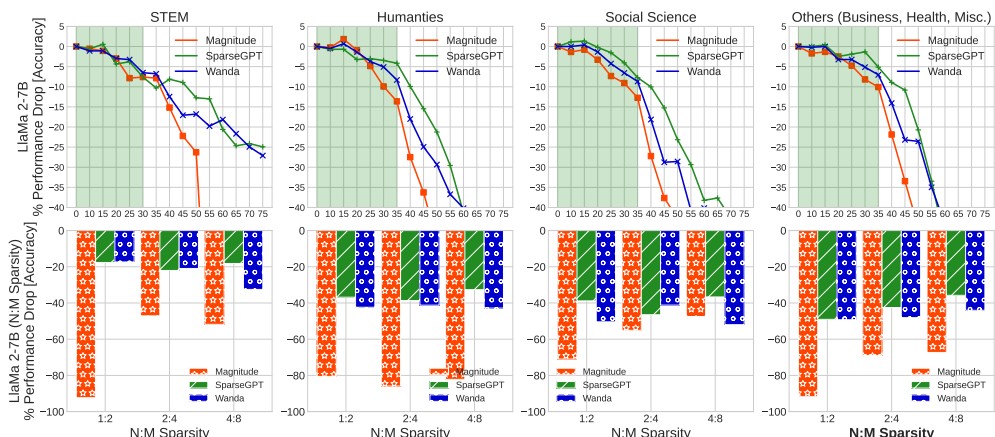

Figure 16: **Compressed LLMs for Multiple-Choice Reasoning based QA.** Performance comparison of compressed LLaMa-2 7B on MCR-QA tasks using the MMLU benchmark (Hendrycks et al., 2020). Results presented are for structured (N:M sparsity) and unstructured sparsity.

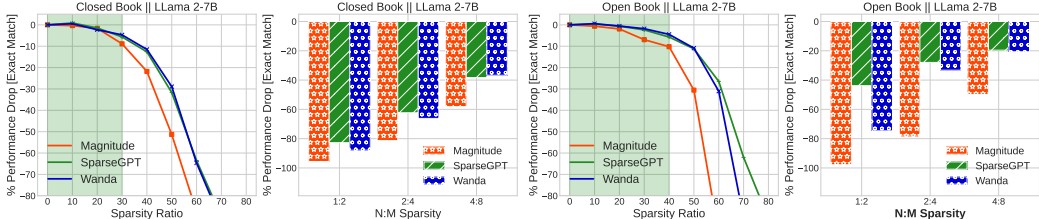

Figure 17: **Compressed LLMs for In-context Retrieval Augmented QA.** Performance comparison of compressed LLaMa-2 7B on ICRA-QA task. We present head-to-head comparison of closed-book evaluation (no external knowledge is augmented in-context) with open-book evaluation (external knowledge is augmented in-context). Results presented are for structured N:M sparsity and unstructured sparsity.

