# OpenReview forum: "Compressing LLMs: The Truth is Rarely Pure and Never Simple"
_ICLR.cc/2024/Conference — ICLR 2024 poster_

### Official Review · Reviewer_X7p4 · 2023-10-18

**Soundness:** 3 good
**Presentation:** 4 excellent
**Contribution:** 2 fair
**Rating:** 6
**Confidence:** 4

**Summary:**

The paper benchmarks compressing techniques, pruning and quantization, on various datasets and metrics, highlighting that a common evaluation metric, perplexity, does not always translate into real-world values.

**Strengths:**

The paper is well-written, all the plots are clear and comprehensive. It's easy to understand information in most of the cases (see weaknesses for improvement).
Authors correctly pinpoint that most of the compression algorithms do not validate extensively on real-world scenarios and show some interesting insights. For example, there is evidence that pruning may not work as well as quantization methods. Or that 8-bit quantization works well on majority on datasets. These insights are easy to use and can have a big impact to how practitioners use these models.

**Weaknesses:**

A few ways to improve this paper.
1. More LLMs. Right now, it's only vicuna model (7B and 13B), having other architectures and perhaps bigger sizes (e.g. llama 70b) would add more evidence for the insights.
2. Add prompt designs to the main body of the paper. This will make it easy to understand how each task is distinct.
3. More quantization methods. AWQ/SmoothQuant are interesting to see.

**Questions:**

Why do we see these performance degradations? Any way to explain why some methods work and some do not?
Are there are any counter forces one can use to mitigate performance degradation while still preserving the quality?

---

> ### Author Response · Authors · 2023-11-18
> **Authors Response to X7p4**
>
> We first like to thank you for your time to review our work. We greatly appreciate that you have found our work to be well-written, comprehensive, and have interesting insights. We are glad that you think our work can have a big impact on how practitioners use these models. For the ease of use, we reiterate that we will be releasing codes, experimental details, and host the datasets on Dropbox, which will help LLM compression researchers use LLM-KICK with ease for evaluating their novel algorithms.
>
> Next, we would like to address all the weaknesses pointed by you point-by-point below:
>
> > **1. More LLMs. Right now, it's only vicuna model (7B and 13B), having other architectures and perhaps bigger sizes (e.g. llama 70b) would add more evidence for the insights.**
>
> Thank you for your suggestion and we strongly agree that extending our experiments beyond Vicuna will bolster our work. In that direction, we have included some additional experiments with LLaMa-2 as you suggested in Appendix A6 which obey our previous findings. For larger model size, we additionally provide some results for Vicuna-33B as follows which again aligns with our conclusion. We also have plans to run our experiments on llama-2 70B as pointed out by you, and include in our final draft (rebuttal duration doesn’t permit us to complete this experiment).
>
> |Factoid-QA|Dense|10%|20%|30%|40%|50%|60%|
> | ------------- |:-------------:|:-------------:|:-------------:|:-------------:|:-------------:|:-------------:|:-------------:|
> |Magnitude|76.92|76.72|71.73|59.44|46.75|42.46|3.10|
> |SparseGPT|76.92|73.93|70.43|61.24|50.85|42.16|29.37|
> |Wanda|76.92|73.63|72.73|65.03|50.65|52.05|39.36|
>
> > **2. Add prompt designs to the main body of the paper. This will make it easy to understand how each task is distinct.**
>
> Thank you for this great suggestion. We agree that the addition of prompt design/example will improve the distinguishability of our task settings. We will surely work towards incorporating this valuable suggestion into our final draft.
>
> > **3. More quantization methods. AWQ/SmoothQuant are interesting to see.**
>
> Thank you for your suggestion. As also pointed out by b73d, we considered evaluating two more quantization methods AWQ and LLM.int8() across our different task settings and we summarize our results on Vicuna-7B as in the following table. We observe that LLM.int8() despite its simplicity and ease-of-use, achieves better results than AWQ (8-bit), and GPTQ (8-bit) across all listed tasks. Note that we couldn’t complete our experiments on summarization and instruction following due to dependency and time constraints (on OpenAI) but we will update them in the final draft.
>
>
> | | GPTQ | AWQ | LLM-int() |
> | ------------- |:-------------:|:-------------:|:-------------:|
> |**Factoid-QA**|60.14%|60.31%|61.02%|
> |**MCR-QA (MMLU)**|47.10%|47.18%|47.82%|
> |**Retrieval Augmented QA**|75.55%|75.89%|75.91%|

---

> > ### Author Response · Authors · 2023-11-18
> > **Authors Response to X7p4 (2/2)**
> >
> > > **4. Why do we see these performance degradations? Any way to explain why some methods work and some do not? Are there are any counter forces one can use to mitigate performance degradation while still preserving the quality?**
> >
> > Thank you for raising this question. We would like to bring attention to Junk DNA Hypotheis (JDH) recently proposed by https://arxiv.org/abs/2310.02277 . According to JDH, small-magnitude weights may appear "useless" for simple tasks and suitable for pruning (in most SoTA pruning methods), but actually encode crucial knowledge necessary for solving more difficult downstream tasks. Removing these seemingly insignificant weights can lead to irreversible knowledge forgetting and performance damage in difficult tasks. Their finding reveals  fresh insights into how LLMs encode knowledge in a task-sensitive manner and pruning them significantly hurt the performance. On the other hand, quantization doesn’t remove these small-magnitude weights completely, which can explain why quantization is more successful than pruning for LLMs. Another recent work, https://arxiv.org/abs/2307.02973 provides a detailed through analysis but wrt small-scale models like ResNet, EfficientNet, etc.
> >
> > To answer the second part of your question, yes, recently there has been some attention towards exploring how to mitigate such performance degradation while still preserving the accuracy. https://arxiv.org/pdf/2310.06927.pdf found that standard loss-based fine-tuning may fail to recover accuracy at high sparsities, but with a type of per-token L2 knowledge distillation, the sparse fine-tuning can reach 75% sparsity without accuracy drops. Similarly, https://arxiv.org/abs/2310.00867 argues that compression does not irretrievably erase LLM model knowledge but displace it, necessitating a new inference path. They propose inference-time dynamic prompting (IDP), a mechanism that autonomously chooses from a set of curated prompts based on the context of each individual input to rewire the inference paths, and suggest prompting might be all you need post-LLM compression. However, note that their evaluation is limited to GSM-8K and it will be interesting to study how their claims translate to LLM-KICK. Not limited to this, some works have also observed the effectiveness of LoRA-based fine-tuning techniques to mitigate performance degradations.
> >
> >
> > *We sincerely hope our responses answer your questions, and we again take this opportunity to thank you for reviewing our work. We would be very happy to respond to any further questions you might have.*

---

### Official Review · Reviewer_b73d · 2023-10-23

**Soundness:** 3 good
**Presentation:** 3 good
**Contribution:** 3 good
**Rating:** 8
**Confidence:** 4

**Summary:**

This paper revisits the efficiency of some compression (pruning and quantization) techniques for LLMs. It conveys that more than perplexity is needed for performance comparison among compressed and uncompressed models. It displays several tasks and benchmarks over which the compressed models exhibit performance degradation despite having similar perplexities.

Essentially, the proposed benchmarks show that quantization if mostly more efficient than pruning where structured pruning appears to offer least ML performance. Interestingly, the paper also shows that even 8-bit quantization if not on par with the uncompressed baseline.

Importantly, the paper states that their related codes are planed to be open-sourced

**Strengths:**

1. Compression of LLMs is very timely and important.

2. The paper reveals new and yet widely unknown gaps in compressed LLMs in comparison to their uncompressed counterparts.

3. The paper shows that compressed models may offer better performance in some tasks (e.g., In-Context Text Summarization) than others (e.g., Factoid-based Question Answering)

4. The authors plan to release their code which may be help in the development of future compression techniques.

**Weaknesses:**

1. It would make the conclusions more robust and convincing if the evaluations use more than a single family of LLMs (i.e., Vicuna).  Why not repeat these experiments with, e.g., Llama 2 and Falcon?

2. Regarding the observation that even 8-bit quantization has evident gaps with respect to uncompressed models, have the authors considered evaluating LLM.int8()?  (https://arxiv.org/pdf/2208.07339.pdf)

3. It would help the reader to have a table summarizing all the tasks' performance over the different architectures, compression techniques and the their resulting perplexity.

**Questions:**

See weakness 1 and 2. Also, does the authors have insights regarding why, in the paper's evaluation, quantization works better than pruning?

---

> ### Author Response · Authors · 2023-11-18
> **Authors Response to b73d**
>
> We would first like to thank you for the time to review our work. We greatly appreciate that you have found our work to be important, timely, and revealing novel widely unknown gaps in compressed LLMs in comparison to their uncompressed counterparts. We again assert that we will be releasing codes, experimental details, and host the datasets on Dropbox, which will help LLM compression researchers use LLM-KICK with ease for evaluating their novel algorithms.
>
> Next, we would like to address all the weaknesses pointed by you point-by-point below:
>
>
> > **1. Extending experiments beyond Vicuna Family?**
>
> Thank you for your suggestion and we strongly agree that extending our experiments beyond a single family of LLMs will bolster our work. In that direction, we have included some additional experiments with LLaMa-2 as you suggested in Appendix A6 (Factoid-QA, MCR-QA, ICR-QA) which obey our previous findings. For larger model size, we additionally provide some results for Vicuna-33B as follows which again aligns with our conclusion. We are running more experiments on larger LLMs which will be included in our final draft.
>
> |Factoid-QA|Dense|10%|20%|30%|40%|50%|60%|
> | ------------- |:-------------:|:-------------:|:-------------:|:-------------:|:-------------:|:-------------:|:-------------:|
> |Magnitude|76.92|76.72|71.73|59.44|46.75|42.46|3.10|
> |SparseGPT|76.92|73.93|70.43|61.24|50.85|42.16|29.37|
> |Wanda|76.92|73.63|72.73|65.03|50.65|52.05|39.36|
>
>
>
> > **2. Comparison with LLM.int8() ?**
>
> Thank you for your suggestion. As also pointed out by X7p4, we considered evaluating AWQ and LLM.int8() across our different task settings and we summarize our results on Vicuna-7B as in the following table. We observe that LLM.int8() despite its simplicity and ease-of-use, achieves better results than AWQ (8-bit), and GPTQ (8-bit) across all listed tasks. Note that we couldn’t complete our experiments on summarization and instruction following due to dependency and time constraints but we will update them in the final draft.
>
> | | GPTQ | AWQ | LLM-int() |
> | ------------- |:-------------:|:-------------:|:-------------:|
> |**Factoid-QA**|60.14%|60.31%|61.02%|
> |**MCR-QA (MMLU)**|47.10%|47.18%|47.82%|
> |**Retrieval Augmented QA**|75.55%|75.89%|75.91%|
>
> > **3. Table summarizing all the tasks' performance?**
>
> Thank you again for your great suggestion and we agree that a table summarizing all task performance will be very helpful for the readers to quickly digest our experimental insights. We promise to include it in our final draft with an additional page.
>
> > **4. Quantization works better than pruning?**
>
> To answer this question, we would like to bring attention to Junk DNA Hypothesis (JDH) recently proposed by https://arxiv.org/abs/2310.02277 . According to JDH, small-magnitude weights may appear "useless" for simple tasks and suitable for pruning (in most SoTA pruning methods), but actually encode crucial knowledge necessary for solving more difficult downstream tasks. Removing these seemingly insignificant weights can lead to irreversible knowledge forgetting and performance damage in difficult tasks. Their finding reveals fresh insights into how LLMs encode knowledge in a task-sensitive manner and pruning them significantly hurt the performance. On the other hand, quantization doesn’t remove these small-magnitude weights completely, which can explain why quantization is more friendly to pruning for LLMs. Another recent work, https://arxiv.org/abs/2307.02973 provides a detailed thorough analysis but wrt small-scale models like ResNet, EfficientNet, etc.
>
> *We sincerely hope our responses answer your questions, and we again would like to take this opportunity to thank you for highly rating our work.*

---

### Official Review · Reviewer_PUep · 2023-10-24

**Soundness:** 3 good
**Presentation:** 3 good
**Contribution:** 3 good
**Rating:** 5
**Confidence:** 4

**Summary:**

This paper benchmarks a few LLM compression methods based on quantization and pruning, on different datasets.

**Strengths:**

I think it is important to have a more fine-grained understanding of compression methods, specially to design new algorithms that can improve upon current weaknesses.

**Weaknesses:**

- This paper is essentially benchmarking a few algorithms on a few datasets. Although the insights are interesting, the paper does not include any new model, data or algorithm, which I'd say makes this paper more suitable for a workshop, not a full conference paper.

- Some arguments are rather subjective. Why choose the 5% threshold? If we change the threshold to 10% it seems 4-bit quantization is then in the range in most cases, and sparse models can still be "competitive" for around 50% sparsity.

- The loss of accuracy also has to be contextualized with the inference time speedups. If a 5% loss of accuracy leads to a 10% reduction in inference time, I'd call that successful.

**Questions:**

The authors say that SparseGPT is data-free. Is that true?

---

> ### Author Response · Authors · 2023-11-18
> **Authors Response to PUep**
>
> We would first like to thank you for the time to review our work. We deeply appreciate that you have found that our work provides a more fine-grained understanding of compression methods. We are glad that you think our work provides interesting insights for SoTA LLM compression methods.
>
> Next, we would like to address all the weaknesses pointed by you point-by-point below:
>
> > **paper does not include any new model, data or algorithm, which I'd say makes this paper more suitable for a workshop, not a full conference paper**
>
> We politely disagree with your comment that without proposing any new model, data, or algorithm, a submitted work is not suitable for a conference paper. To provide more context, [1,2,3,4,5,6,7,8] are some among numerous examples of top full conference papers (some cited >150) which doesn’t propose any new model, data, or algorithm. We kindly request you to acknowledge that scientific research can encompass novelty in various forms. We believe that thoroughly exploring existing solutions, and providing novel insights holds equal significance to the pursuit of new methods or architectures. We also very politely request you to look at other reviewers who find our work very important, timely, and can have a big impact on the LLM compression community.
>
> [1] NeurIPS’19. A Meta-Analysis of Overfitting in Machine Learning, https://papers.nips.cc/paper_files/paper/2019/hash/ee39e503b6bedf0c98c388b7e8589aca-Abstract.html
>
> [2] NeurIPS’20. The Lottery Ticket Hypothesis for Pre-trained BERT Networks https://proceedings.neurips.cc/paper/2020/file/b6af2c9703f203a2794be03d443af2e3-Paper.pdf
>
> [3] CVPR’21. The Lottery Tickets Hypothesis for Supervised and Self-supervised Pre-training in Computer Vision Models https://openaccess.thecvf.com/content/CVPR2021/papers/Chen_The_Lottery_Tickets_Hypothesis_for_Supervised_and_Self-Supervised_Pre-Training_in_CVPR_2021_paper.pdf
>
> [4] CVPR’23. Reproducible scaling laws for contrastive language-image learning, https://openaccess.thecvf.com/content/CVPR2023/papers/Cherti_Reproducible_Scaling_Laws_for_Contrastive_Language-Image_Learning_CVPR_2023_paper.pdf
>
> [5] ICLR’23 - Top25%: Sparsity May Cry: Let Us Fail (Current) Sparse Neural Networks Together! https://openreview.net/forum?id=J6F3lLg4Kdp
>
> [6] ICLR’22: The Unreasonable Effectiveness of Random Pruning: Return of the Most Naive Baseline for Sparse Training https://arxiv.org/pdf/2202.02643.pdf
>
> [7] NeurIPS’22: Subgroup Robustness Grows On Trees: An Empirical Baseline Investigation https://proceedings.neurips.cc/paper_files/paper/2022/file/408cf1a1d9ff35d5fea7075565dbf434-Paper-Conference.pdf
>
> [8] ACL’23: Understanding Factual Errors in Summarization: Errors, Summarizers, Datasets, Error Detectors https://aclanthology.org/2023.acl-long.650/
>
> > **Some arguments are rather subjective. Why choose the 5% threshold? If we change the threshold to 10% it seems 4-bit quantization is then in the range in most cases, and sparse models can still be "competitive" for around 50% sparsity.**
>
> Thank you for bringing this up and we are very happy to clarify your doubts. Firstly, **our threshold is simply an indicator to illustrate the tolerance level of performance drop** when we start compressing any LLM. Many prior works [1,2,3] consider matching thresholds to be the same as the dense subnetwork performance or within the margins of 1%. However, in our work, we carefully relaxed it to 5% performance drop as an acceptable tolerance (before calling the compressed model useless) keeping in mind that the performance of compressed LLM on any of our task categories/disciplines remains above the random guess. For example, to provide context, as you have mentioned, 10% threshold on MMLU can leave performance on some disciplines like Algebra, Econometrics drop to 19.7%, 21.1% which is below the random guess (25%) given MMLU has one correct out of four choices. Again, note that this threshold has nothing to do with any performance or insight reported in our draft. We will add additional clarification in our final draft.
>
>
> [1] CVPR’21: The Lottery Tickets Hypothesis for Supervised and Self-supervised Pre-training in Computer Vision Models
>
> [2] NeurIPS’23: The Emergence of Essential Sparsity in Large Pre-trained Models: The Weights that Matter
>
> [3] NeurIPS’20: The Lottery Ticket Hypothesis for Pre-trained BERT Networks

---

> > ### Author Response · Authors · 2023-11-18
> > **Authors Response to PUep (2/2)**
> >
> > > **The loss of accuracy also has to be contextualized with the inference time speedups. If a 5% loss of accuracy leads to a 10% reduction in inference time, I'd call that successful.**
> >
> > Thank you for your comment. However, we want to clarify that for our work, inference time speedups are out-of-scope and we focus on the performance perspective of compression on LLMs. Speedup measurement of autoregressive LLMs depends on a variety of factors like task setting (how many tokens are generated for summarization or conversations, vs Multiple-choice where only True/False have to be selected), GPU architectures, how weights are loaded (on one GPU or shredded across multiple), among many others. Note that our work is not about calling if a specific method is successful or not; rather exploring how compressed LLMs behave in various task settings to provide a careful, thorough understanding of the merits/plights of compressed LLMs, e.g., even highly compressed LLMs are extremely good at in-context summarization/retrieval-augmented QA despite their performance significantly suffering when they have to access their own knowledge for factoid-based QA.
> >
> > > **The authors say that SparseGPT is data-free. Is that true?**
> >
> > Many thanks for pointing this out. We will correct it as data-free -> downstream data-free which means no downstream data is used to prune the models, rather only a small amount of calibration data is used for pruning.
> >
> > *We sincerely hope our responses answer your questions, and we again take this opportunity to thank you for reviewing our work. We would be very happy to respond to any further questions you might have.*

---

> > > ### Comment · Reviewer_PUep · 2023-11-21
> > >
> > > Thank you for your response.
> > >
> > > - I think your point about certain tasks being worse than random makes sense. I'm not sure if this is clarified in the paper, but it would be good to discuss such objectively bad cases, rather than focusing on some threshold.
> > >
> > > - I still think inference time is important and should be contained in the comparisons.
> > >
> > > Based on this, I will increase my score to 5, though I think without inference time the paper still lacks.

---

> > > > ### Author Response · Authors · 2023-11-21
> > > > **Response to Reviewer PUep**
> > > >
> > > > *We are glad that our arguments have clarified some of your concerns. We also like to thank you for increasing your rating for our work following that. We are also sincerely considering your suggestion to conduct some inference time statistics, and planning to incorporate them in our final version.*

---

### Official Review · Reviewer_QryF · 2023-11-04

**Soundness:** 3 good
**Presentation:** 3 good
**Contribution:** 3 good
**Rating:** 8
**Confidence:** 4

**Summary:**

The paper is very timely and identifies an important gap in evaluation of compression on LLMs. It points out how perplexity is not a correct metric to evaluate compression benchmarks (which is also previously observed in other contexts). They curate a set of datasets which can form a better representation of language model capabilities.

**Strengths:**

- timely ... with an array of papers on compressing LLMs with especially surprising results such as training free pruning coming out. It is important to enable researchers with better tools of evaluation
- provides a decent array of dataset benchmarks that will be use ful in research.
- clearly shows the gap between evaluation of perplexity and other proposed datasets.

**Weaknesses:**

Not weaknesses. but suggestions.
1. add a summarizing table to list dataset statistics.

**Questions:**

None

---

> ### Author Response · Authors · 2023-11-18
> **Author Response to Reviewer QryF**
>
> We would first like to thank you for the time to review our work. We greatly appreciate that you have found our work to be important, timely, and significantly necessary considering the growing interest in LLM compression, and the lack of any standard evaluation protocol to elucidate the merits and plights of novel algorithms.
>
> We also significantly value your suggestion to include the dataset statistics, and we will update our Appendix A5 to provide more details. We also have plans to release codes, experimental details, and host the datasets on Dropbox, which will help LLM compression researchers use LLM-KICK with ease for evaluating their novel algorithms.
>
> We again take this opportunity to thank you for highly rating our work.

---

### Meta-Review · Area_Chair_9tvQ · 2023-12-05

**Metareview:**

The paper makes significant scientific claims about the inadequacy of perplexity as a sole metric for evaluating LLM compression and introduces more comprehensive benchmarks and datasets for a more accurate assessment. The findings indicate a nuanced understanding of how different compression techniques, specifically quantization and pruning, affect the performance of large language models.

**Justification For Why Not Higher Score:**

The authors have effectively addressed the primary concerns raised by the reviewers. However, there are still some aspects that require further clarification in the final version of the paper. For instance, the observation regarding certain tasks performing worse than random is intriguing and warrants a more detailed discussion. It would be beneficial if the paper could elaborate on these objectively poor-performing cases, instead of concentrating solely on a specific performance threshold. Additionally, the importance of inference time as a factor in the comparisons should not be overlooked and deserves inclusion in the final analysis.

**Justification For Why Not Lower Score:**

The paper addresses a significant issue in the evaluation of large language models (LLMs) in the context of compression techniques, backed by comprehensive experimental studies.

---

### Decision · Program_Chairs · 2024-01-16

Accept (poster)